# Beyond Word Importance: Contextual Decomposition to Extract Interactions from LSTMs

**W. James Murdoch** [*]
Department of Statistics
University of California, Berkeley
jmurdoch@berkeley.edu

**Peter J. Liu**
Google Brain
Mountain View, CA

**Bin Yu**
Department of Statistics
Department of EECS
University of California, Berkeley

## Abstract

The driving force behind the recent success of LSTMs has been their ability to learn complex and non-linear relationships. Consequently, our inability to describe these relationships has led to LSTMs being characterized as black boxes. To this end, we introduce contextual decomposition (CD), an interpretation algorithm for analysing individual predictions made by standard LSTMs, without any changes to the underlying model. By decomposing the output of a LSTM, CD captures the contributions of combinations of words or variables to the final prediction of an LSTM. On the task of sentiment analysis with the Yelp and SST data sets, we show that CD is able to reliably identify words and phrases of contrasting sentiment, and how they are combined to yield the LSTM's final prediction. Using the phrase-level labels in SST, we also demonstrate that CD is able to successfully extract positive and negative negations from an LSTM, something which has not previously been done.

## 1 Introduction

In comparison with simpler linear models, techniques from deep learning have achieved impressive accuracy by effectively learning non-linear interactions between features. However, due to our inability to describe the learned interactions, this improvement in accuracy has come at the cost of state of the art predictive algorithms being commonly regarded as black-boxes. In the domain of natural language processing (NLP), Long Short Term Memory networks (LSTMs) (Hochreiter & Schmidhuber, 1997) have become a basic building block, yielding excellent performance across a wide variety of tasks (Sutskever et al., 2014) (Rajpurkar et al., 2016) (Melis et al., 2017), while remaining largely inscrutable.

In this work, we introduce contextual decomposition (CD), a novel interpretation method for explaining individual predictions made by an LSTM without any modifications to the underlying model. CD extracts information about not only which words contributed to a LSTM's prediction, but also how they were combined in order to yield the final prediction. By mathematically decomposing the LSTM's output, we are able to disambiguate the contributions made at each step by different parts of the sentence.

To validate the CD interpretations extracted from an LSTM, we evaluate on the problem of sentiment analysis. In particular, we demonstrate that CD is capable of identifying words and phrases of differing sentiment within a given review. CD is also used to successfully extract positive and negative negations from an LSTM, something that has not previously been done. As a consequence of this analysis, we also show that prior interpretation methods produce scores which have document-level information built into them in complex, unspecified ways. For instance, prior work often identifies strongly negative phrases contained within positive reviews as neutral, or even positive.

---

[*]Work started during internship at Google Brain

## 2 RELATED WORK

The most relevant prior work on interpreting LSTMs has focused on approaches for computing word-level importance scores, with evaluation protocols varying greatly. Murdoch & Szlam (2017) introduced a decomposition of the LSTM's output embedding into a sum over word coefficients, and demonstrated that those coefficients are meaningful by using them to distill LSTMs into rules-based classifiers. Li et al. (2016) took a more black box approach, called Leave One Out, by observing the change in log probability resulting from replacing a given word vector with a zero vector, and relied solely on anecdotal evaluation. Finally, Sundararajan et al. (2017) presents a general gradient-based technique, called Integrated Gradients, which was validated both theoretically and with empirical anecdotes. In contrast to our proposed method, this line of work has been limited to word-based importance scores, ignoring the interactions between variables which make LSTMs so accurate.

Another line of work (Karpathy et al., 2015) (Strobelt et al., 2016) has focused on analysing the movement of raw gate activations over a sequence. Karpathy et al. (2015) was able to identify some co-ordinates of the cell state that correspond to semantically meaningful attributes, such as whether the text is in quotes. However, most of the cell co-ordinates were uninterpretable, and it is not clear how these co-ordinates combine to contribute to the actual prediction.

Decomposition-based approaches to interpretation have also been applied to convolutional neural networks (CNNs) (Bach et al., 2015) (Shrikumar et al., 2017). However, they have been limited to producing pixel-level importance scores, ignoring interactions between pixels, which are clearly quite important. Our approach is similar to these in that it computes an exact decomposition, but we leverage the unique gating structure of LSTMs in order to extract interactions.

Attention based models (Bahdanau et al., 2014) offer another means of providing some interpretability. Such models have been successfully applied to many problems, yielding improved performance (Rush et al., 2015) (Xu et al., 2015). In contrast to other word importance scores, attention is limited in that it only provides an indirect indicator of importance, with no directionality, i.e. what class the word is important for. Although attention weights are often cited anecdotally, they have not been evaluated, empirically or otherwise, as an interpretation technique. As with other prior work, attention is also incapable of describing interactions between words.

## 3 CONTEXTUAL DECOMPOSITION OF LSTMS

Given an arbitrary phrase contained within an input, we present a novel decomposition of the output of an LSTM into a sum of two contributions: those resulting solely from the given phrase, and those involving other factors. The key insight behind this decomposition is that the gating dynamics unique to LSTMs are a vehicle for modeling interactions between variables.

### 3.1 LONG SHORT TERM MEMORY NETWORKS

Over the past few years, LSTMs have become a core component of neural NLP systems. Given a sequence of word embeddings $x_1, ..., x_T \in \mathbb{R}^{d_1}$, a cell and state vector $c_t, h_t \in \mathbb{R}^{d_2}$ are computed for each element by iteratively applying the below equations, with initialization $h_0 = c_0 = 0$.

$$o_t = \sigma(W_o x_t + V_o h_{t-1} + b_o) \tag{1}$$
$$f_t = \sigma(W_f x_t + V_f h_{t-1} + b_f) \tag{2}$$
$$i_t = \sigma(W_i x_t + V_i h_{t-1} + b_i) \tag{3}$$
$$g_t = \tanh(W_g x_t + V_g h_{t-1} + b_g) \tag{4}$$
$$c_t = f_t \odot c_{t-1} + i_t \odot g_t \tag{5}$$
$$h_t = o_t \odot \tanh(c_t) \tag{6}$$

Where $W_o, W_i, W_f, W_g \in \mathbb{R}^{d_1 \times d_2}$, $V_o, V_f, V_i, V_g \in \mathbb{R}^{d_2 \times d_2}$, $b_o, b_g, b_i, b_g \in \mathbb{R}^{d_2}$ and $\odot$ denotes element-wise multiplication. $o_t, f_t$ and $i_t$ are often referred to as output, forget and input gates, respectively, due to the fact that their values are bounded between 0 and 1, and that they are used in element-wise multiplication.

After processing the full sequence, the final state $h_T$ is treated as a vector of learned features, and used as input to a multinomial logistic regression, often called SoftMax, to return a probability distribution $p$ over $C$ classes, with

$$p_j = \text{SoftMax}(Wh_T)_j = \frac{\exp(W_j h_T)}{\sum_{k=1}^{C} \exp(W_k h_t)} \tag{7}$$

## 3.2 Contextual Decomposition of LSTM

We now introduce contextual decomposition, our proposed method for interpreting LSTMs. Given an arbitrary phrase $x_q, ..., x_r$, where $1 \le q \le r \le T$, we now decompose each output and cell state $c_t, h_t$ in Equations 5 and 6 into a sum of two contributions.

$$h_t = \beta_t + \gamma_t \tag{8}$$
$$c_t = \beta_t^c + \gamma_t^c \tag{9}$$

The decomposition is constructed so that $\beta_t$ corresponds to contributions made solely by the given phrase to $h_t$, and that $\gamma_t$ corresponds to contributions involving, at least in part, elements outside of the phrase. $\beta_t^c$ and $\gamma_t^c$ represent analogous contributions to $c_t$.

Using this decomposition for the final output state $Wh_T$ in Equation 7 yields

$$p = \text{SoftMax}(W\beta_T + W\gamma_T) \tag{10}$$

Here $W\beta_T$ provides a quantitative score for the phrase's contribution to the LSTM's prediction. As this score corresponds to the input to a logistic regression, it may be interpreted in the same way as a standard logistic regression coefficient.

### 3.2.1 Disambiguating interactions between gates

In the cell update Equation 5, neuron values in each of $i_t$ and $g_t$ are independently determined by both the contribution at that step, $x_t$, as well as prior context provided by $h_{t-1} = \beta_{t-1} + \gamma_{t-1}$. Thus, in computing the element-wise product $i_t \odot g_t$, often referred to as gating, contributions made by $x_t$ to $i_t$ interact with contributions made by $h_t$ to $g_t$, and vice versa.

We leverage this simple insight to construct our decomposition. First, assume that we have a way of linearizing the gates and updates in Equations 2, 3, 4 so that we can write each of them as a linear sum of contributions from each of their inputs.

$$i_t = \sigma(W_i x_t + V_i h_{t-1} + b_i) \tag{11}$$
$$= L_\sigma(W_i x_t) + L_\sigma(V_i h_{t-1}) + L_\sigma(b_i) \tag{12}$$

When we use this linearization in the cell update Equation 5, the products between gates become products over linear sums of contributions from different factors. Upon expanding these products, the resulting cross-terms yield a natural interpretation as being interactions between variables. In particular, cross-terms can be assigned as to whether they resulted solely from the phrase, e.g. $L_\sigma(V_i \beta_{t-1}) \odot L_{\tanh}(V_g \beta_{t-1})$, from some interaction between the phrase and other factors, e.g. $L_\sigma(V_i \beta_{t-1}) \odot L_{\tanh}(V_g \gamma_{t-1})$, or purely from other factors, e.g. $L_\sigma(b_i) \odot L_{\tanh}(V_g \gamma_{t-1})$.

Mirroring the recurrent nature of LSTMs, the above insights allow us to recursively compute our decomposition, with the initializations $\beta_0 = \beta_0^c = \gamma_0 = \gamma_0^c = 0$. We derive below the update equations for the case where $q \le t \le r$, so that the current time step is contained within the phrase. The other case is similar, and the general recursion formula is provided in Appendix 6.2.

For clarity, we decompose the two products in the cell update Equation 5 separately. As discussed above, we simply linearize the gates involved, expand the resulting product of sums, and group the cross-terms according to whether or not their contributions derive solely from the specified phrase, or

otherwise. Terms are determined to derive solely from the specified phrase if they involve products from some combination of $\beta_{t-1}, \beta_{t-1}^c, x_t$ and $b_i$ or $b_g$ (but not both). When $t$ is not within the phrase, products involving $x_t$ are treated as not deriving from the phrase.

$$
\begin{align}
f_t \odot c_{t-1} &= (L_\sigma(W_f x_t) + L_\sigma(V_f \beta_{t-1}) + L_\sigma(V_f \gamma_{t-1}) + L_\sigma(b_f)) \odot (\beta_{t-1}^c + \gamma_{t-1}^c) \tag{13} \\
&= ([L_\sigma(W_f x_t) + L_\sigma(V_f \beta_{t-1}) + L_\sigma(b_f)] \odot \beta_{t-1}^c) \tag{14} \\
&\quad + (L_\sigma(V_f \gamma_{t-1}) \odot \beta_{t-1}^c + f_t \odot \gamma_{t-1}^c) \\
&= \beta_t^f + \gamma_t^f \tag{15}
\end{align}
$$

$$
\begin{align}
i_t \odot g_t &= [L_\sigma(W_i x_t) + L_\sigma(V_i \beta_{t-1}) + L_\sigma(V_i \gamma_{t-1}) + L_\sigma(b_i)] \tag{16} \\
&\quad \odot [L_{\tanh}(W_g x_t) + L_{\tanh}(V_g \beta_{t-1}) + L_{\tanh}(V_g \gamma_{t-1}) + L_{\tanh}(b_g)] \\
&= [L_\sigma(W_i x_t) \odot [L_{\tanh}(W_g x_t) + L_{\tanh}(V_g \beta_{t-1}) + L_{\tanh}(b_g)] \tag{17} \\
&\quad + L_\sigma(V_i \beta_{t-1}) \odot [L_{\tanh}(W_g x_t) + L_{\tanh}(V_g \beta_{t-1}) + L_{\tanh}(b_g)] \\
&\quad + L_\sigma(b_i) \odot [L_{\tanh}(W_g x_t) + L_{\tanh}(V_g \beta_{t-1})]] \\
&\quad + [L_\sigma(V_i \gamma_{t-1}) \odot g_t + i_t \odot L_{\tanh}(V_g \gamma_{t-1}) - L_\sigma(V_i \gamma_{t-1}) \odot L_{\tanh}(V_g \gamma_{t-1}) \\
&\quad + L_\sigma(b_i) \odot L_{\tanh}(b_g)] \\
&= \beta_t^u + \gamma_t^u \tag{18}
\end{align}
$$

Having decomposed the two components of the cell update equation, we can attain our decomposition of $c_t$ by summing the two contributions.

$$
\begin{align}
\beta_t^c &= \beta_t^f + \beta_t^u \tag{19} \\
\gamma_t^c &= \gamma_t^f + \gamma_t^u \tag{20}
\end{align}
$$

Once we have computed the decomposition of $c_t$, it is relatively simple to compute the resulting transformation of $h_t$ by linearizing the $\tanh$ function in 6. Note that we could similarly decompose the output gate as we treated the forget gate above, but we empirically found this to not produce improved results.

$$
\begin{align}
h_t &= o_t \odot \tanh(c_t) \tag{21} \\
&= o_t \odot [L_{\tanh}(\beta_t^c) + L_{\tanh}(\gamma_t^c)] \tag{22} \\
&= o_t \odot L_{\tanh}(\beta_t^c) + o_t \odot L_{\tanh}(\gamma_t^c) \tag{23} \\
&= \beta_t + \gamma_t \tag{24}
\end{align}
$$

### 3.2.2 LINEARIZING ACTIVATION FUNCTIONS

We now describe the linearizing functions $L_\sigma, L_{\tanh}$ used in the above decomposition. Formally, for arbitrary $\{y_1, ..., y_N\} \in \mathbb{R}$, where $N \leq 4$, the problem is how to write

$$
\tanh(\sum_{i=1}^{N} y_i) = \sum_{i=1}^{N} L_{\tanh}(y_i) \tag{25}
$$

In the cases where there is a natural ordering to $\{y_i\}$, prior work (Murdoch & Szlam, 2017) has used a telescoping sum consisting of differences of partial sums as a linearization technique, which we show below.

$$
L'_{\tanh}(y_k) = \tanh(\sum_{j=1}^{k} y_j) - \tanh(\sum_{j=1}^{k-1} y_j) \tag{26}
$$

However, in our setting $\{y_i\}$ contains terms such as $\beta_{t-1}$, $\gamma_{t-1}$ and $x_t$, which have no clear ordering. Thus, there is no natural way to order the sum in Equation 26. Instead, we compute an average over all orderings. Letting $\pi_1, ..., \pi_{M_N}$ denote the set of all permutations of $1, ..., N$, our score is given below. Note that when $\pi_i(j) = j$, the corresponding term is equal to equation 26.

$$L_{\tanh}(y_k) = \frac{1}{M_N} \sum_{i=1}^{M_N} [\tanh(\sum_{j=1}^{\pi_i^{-1}(k)} y_{\pi_i(j)}) - \tanh(\sum_{j=1}^{\pi_i^{-1}(k)-1} y_{\pi_i(j)})] \tag{27}$$

$L_\sigma$ can be analogously derived. When one of the terms in the decomposition is a bias, we saw improvements when restricting to permutations where the bias is the first term.

As $N$ only ranges between 2 and 4, this linearization generally takes very simple forms. For instance, when $N = 2$, the contribution assigned to $y_1$ is

$$L_{\tanh}(y_1) = \frac{1}{2}([\tanh(y_1) - \tanh(0)] + [\tanh(y_2 + y_1) - \tanh(y_1)]) \tag{28}$$

This linearization was presented in a scalar context where $y_i \in \mathbb{R}$, but trivially generalizes to the vector setting $y_i \in \mathbb{R}^{d_2}$. It can also be viewed as an approximation to Shapely values, as discussed in Lundberg & Lee (2016) and Shrikumar et al. (2017).

## 4    EXPERIMENTS

We now describe our empirical validation of CD on the task of sentiment analysis. First, we verify that, on the standard problem of word-level importance scores, CD compares favorably to prior work. Then we examine the behavior of CD for word and phrase level importance in situations involving compositionality, showing that CD is able to capture the composition of phrases of differing sentiment. Finally, we show that CD is capable of extracting instances of positive and negative negation. Code for computing CD scores is available online [1].

### 4.1    TRAINING DETAILS

We first describe the process for fitting models which are used to produce interpretations. As the primary intent of this paper is not predictive accuracy, we used standard best practices without much tuning. We implemented all models in Torch using default hyperparameters for weight initializations. All models were optimized using Adam (Kingma & Ba, 2014) with the default learning rate of 0.001 using early stopping on the validation set. For the linear model, we used a bag of vectors model, where we sum pre-trained Glove vectors (Pennington et al., 2014) and add an additional linear layer from the word embedding dimension, 300, to the number of classes, 2. We fine tuned both the word vectors and linear parameters. We will use the two data sets described below to validate our new CD method.

#### 4.1.1    STANFORD SENTIMENT TREEBANK

We trained an LSTM model on the binary version of the Stanford Sentiment Treebank (SST) (Socher et al., 2013), a standard NLP benchmark which consists of movie reviews ranging from 2 to 52 words long. In addition to review-level labels, it also provides labels for each phrase in the binarized constituency parse tree. Following the hyperparameter choices in Tai et al. (2015), the word and hidden representations of our LSTM were set to 300 and 168, and word vectors were initialized to pretrained Glove vectors (Pennington et al., 2014). Our LSTM attains 87.2% accuracy, and we also train a logistic regression model with bag of words features, which attains 83.2% accuracy.

---

[1]https://github.com/jamie-murdoch/ContextualDecomposition

### 4.1.2 YELP POLARITY

Originally introduced in Zhang et al. (2015), the Yelp review polarity dataset was obtained from the Yelp Dataset Challenge and has train and test sets of sizes 560,000 and 38,000. The task is binary prediction for whether the review is positive (four or five stars) or negative (one or two stars). The reviews are relatively long, with an average length of 160.1 words. Following the guidelines from Zhang et al. (2015), we implement an LSTM model which attains 4.6% error, and an ngram logistic regression model, which attains 5.7% error. For computational reasons, we report interpretation results on a random subset of sentences of length at most 40 words. When computing integrated gradient scores, we found that numerical issues produced unusable outputs for roughly 6% of the samples. These reviews are excluded.

### 4.1.3 INTERPRETATION BASELINES

We compare the interpretations produced by CD against four state of the art baselines: cell decomposition (Murdoch & Szlam, 2017), integrated gradients (Sundararajan et al., 2017), leave one out (Li et al., 2016), and gradient times input. We refer the reader to Section 2 for descriptions of these algorithms. For our gradient baseline, we compute the gradient of the output probability with respect to the word embeddings, and report the dot product between the word vector and its gradient. For integrated gradients, producing reasonable values required extended experimentation and communication with the creators regarding the choice of baselines and scaling issues. We ultimately used sequences of periods for our baselines, and rescaled the scores for each review by the standard deviation of the scores for that review, a trick not previously mentioned in the literature. To obtain phrase scores for word-based baselines integrated gradients, cell decomposition, and gradients, we sum the scores of the words contained within the phrase.

### 4.2 UNIGRAM (WORD) SCORES

Before examining the novel, phrase-level dynamics of CD, we first verify that it compares favorably to prior work for the standard use case of producing unigram coefficients. When sufficiently accurate in terms of prediction, logistic regression coefficients are generally treated as a gold standard for interpretability. In particular, when applied to sentiment analysis the ordering of words given by their coefficient value provides a qualitatively sensible measure of importance. Thus, when determining the validity of coefficients extracted from an LSTM, we should expect there to be a meaningful relationship between the CD scores and logistic regression coefficients.

In order to evaluate the word-level coefficients extracted by the CD method, we construct scatter plots with each point consisting of a single word in the validation set. The two values plotted correspond to the coefficient from logistic regression and importance score extracted from the LSTM. For a quantitative measure of accuracy, we use pearson correlation coefficient.

We report quantitative and qualitative results in Appendix 6.1.3. For SST, CD and integrated gradients, with correlations of 0.76 and 0.72, respectively, are substantially better than other methods, with correlations of at most 0.51. On Yelp, the gap is not as big, but CD is still very competitive, having correlation 0.52 with other methods ranging from 0.34 to 0.56. Having verified reasonably strong results in this base case, we now proceed to show the benefits of CD.

### 4.3 IDENTIFYING DISSENTING SUBPHRASES

We now show that, for phrases of at most five words, existing methods are unable to recognize subphrases with differing sentiments. For example, consider the phrase "used to be my favorite", which is of negative sentiment. The word "favorite", however, is strongly positive, having a logistic regression coefficient in the 93rd percentile. Nonetheless, existing methods consistently rank "favorite" as being highly negative or neutral. In contrast, as shown in Table 1, CD is able to identify "my favorite" as being strongly positive, and "used to be" as strongly negative. A similar dynamic also occurs with the phrase "not worth the time". The main justification for using LSTMs over simpler models is precisely that they are able to capture these kinds of interactions. Thus, it is important that an interpretation algorithm is able to properly uncover how the interactions are being handled.

| Attribution Method | Heat Map | | | | | | | | |
|---|---|---|---|---|---|---|---|---|---|
| Gradient | used | to | be | my | favorite | not | worth | the | time |
| Leave One Out (Li et al., 2016) | used | to | be | my | favorite | not | worth | the | time |
| Cell decomposition (Murdoch & Szlam, 2017) | used | to | be | my | favorite | not | worth | the | time |
| Integrated gradients (Sundararajan et al., 2017) | used | to | be | my | favorite | not | worth | the | time |
| Contextual decomposition | used | to | be | my | favorite | not | worth | the | time |

Legend: Very Negative | Negative | Neutral | Positive | Very Positive

Table 1: Heat maps for portion of yelp review with different attribution techniques. Only CD captures that "favorite" is positive.

Using the above as a motivating example, we now show that a similar trend holds throughout the Yelp polarity dataset. In particular, we conduct a search for situations similar to the above, where a strongly positive/negative phrase contains a strongly dissenting subphrase. Phrases are scored using the logistic regression with n-gram features described in Section 4.1, and included if their absolute score is over 1.5. We then examine the distribution of scores for the dissenting subphrases, which are analogous to "favorite".

For an effective interpretation algorithm, the distribution of scores for positive and negative dissenting subphrases should be significantly separate, with positive subphrases having positive scores, and vice versa. However, as can be seen in Appendix 6.1.1, for prior methods these two distributions are nearly identical. The CD distributions, on the other hand, are significantly separate, indicating that what we observed anecdotally above holds in a more general setting.

## 4.4 EXAMINING HIGH-LEVEL COMPOSITIONALITY

We now show that prior methods struggle to identify cases where a sizable portion of a review (between one and two thirds) has polarity different from the LSTM's prediction. For instance, consider the review in Table 2, where the first phrase is clearly positive, but the second phrase causes the review to ultimately be negative. CD is the only method able to accurately capture this dynamic.

By leveraging the phrase-level labels provided in SST, we can show that this pattern holds in the general case. In particular, we conduct a search for reviews similar to the above example. The search criteria are whether a review contains a phrase labeled by SST to be of opposing sentiment to the review-level SST label, and is between one and two thirds the length of the review.

In Appendix 6.1.2, we show the distribution of the resulting positive and negative phrases for different attribution methods. A successful interpretation method would have a sizable gap between these two distributions, with positive phrases having mostly positive scores, and negative phrases mostly negative. However, prior methods struggle to satisfy these criteria. 87% of all positive phrases are labelled as negative by integrated gradients, and cell decompositions (Murdoch & Szlam, 2017) even have the distributions flipped, with negative phrases yielding more positive scores than the positive phrases. CD, on the other hand, provides a very clear difference in distributions. To quantify this separation between positive and negative distributions, we examine a two-sample Kolmogorov-Smirnov one-sided test statistic, a common test for the difference of distributions with values ranging from 0 to 1. CD produces a score of 0.74, indicating a strong difference between positive and negative distributions, with other methods achieving scores of 0 (cell decomposition), 0.33 (integrated gradients), 0.58 (leave one out) and 0.61 (gradient), indicating weaker distributional differences. Given that gradient and leave one out were the weakest performers in unigram scores, this provides strong evidence for the superiority of CD.

| Attribution Method | Heat Map |
|---|---|
| Gradient | It's easy to love Robin Tunney – she's pretty and she can act – but it gets harder and harder to understand her choices. |
| Leave one out (Li et al., 2016) | It's easy to love Robin Tunney – she's pretty and she can act – but it gets harder and harder to understand her choices. |
| Cell decomposition (Murdoch & Szlam, 2017) | It's easy to love Robin Tunney – she's pretty and she can act – but it gets harder and harder to understand her choices. |
| Integrated gradients (Sundararajan et al., 2017) | It's easy to love Robin Tunney – she's pretty and she can act – but it gets harder and harder to understand her choices. |
| Contextual decomposition | It's easy to love Robin Tunney – she's pretty and she can act – but it gets harder and harder to understand her choices. |

Legend: Very Negative | Negative | Neutral | Positive | Very Positive

Table 2: Heat maps for portion of review from SST with different attribution techniques. Only CD captures that the first phrase is positive.

## 4.5 CONTEXTUAL DECOMPOSITION (CD) CAPTURES NEGATION

In order to understand an LSTM's prediction mechanism, it is important to understand not just the contribution of a phrase, but how that contribution is computed. For phrases involving negation, we now demonstrate that we can use CD to empirically show that our LSTM learns a negation mechanism.

Using the phrase labels in SST, we search over the training set for instances of negation. In particular, we search for phrases of length less than ten with the first child containing a negation phrase (such as "not" or "lacks", full list provided in Appendix 6.3) in the first two words, and the second child having positive or negative sentiment. Due to noise in the labels, we also included phrases where the entire phrase was non-neutral, and the second child contained a non-neutral phrase. We identify both positive negation, such as "isn't a bad film", and negative negation, such as "isn't very interesting", where the direction is given by the SST-provided label of the phrase.

For a given negation phrase, we extract a negation interaction by computing the CD score of the entire phrase and subtracting the CD scores of the phrase being negated and the negation term itself. The resulting score can be interpreted as an n-gram feature. Note that, of the methods we compare against, only leave one out is capable of producing such interaction scores. For reference, we also provide the distribution of all interactions for phrases of length less than 5.

We present the distribution of extracted scores in Figure 1. For CD, we can see that there is a clear distinction between positive and negative negations, and that the negation interactions are centered on the outer edges of the distribution of interactions. Leave one out is able to capture some of the interactions, but has a noticeable overlap between positive and negative negations around zero, indicating a high rate of false negatives.

## 4.6 IDENTIFYING SIMILAR PHRASES

Another benefit of using CDs for interpretation is that, in addition to providing importance scores, it also provides dense embeddings for arbitrary phrases and interactions, in the form of $\beta_T$ discussed in Section 3.2. We anecdotally show that similarity in this embedding space corresponds to semantic similarity in the context of sentiment analysis.

In particular, for all words and binary interactions, we compute the average embedding $\beta_T$ produced by CD across the training and validation sets. In Table 3, we show the nearest neighbours using a

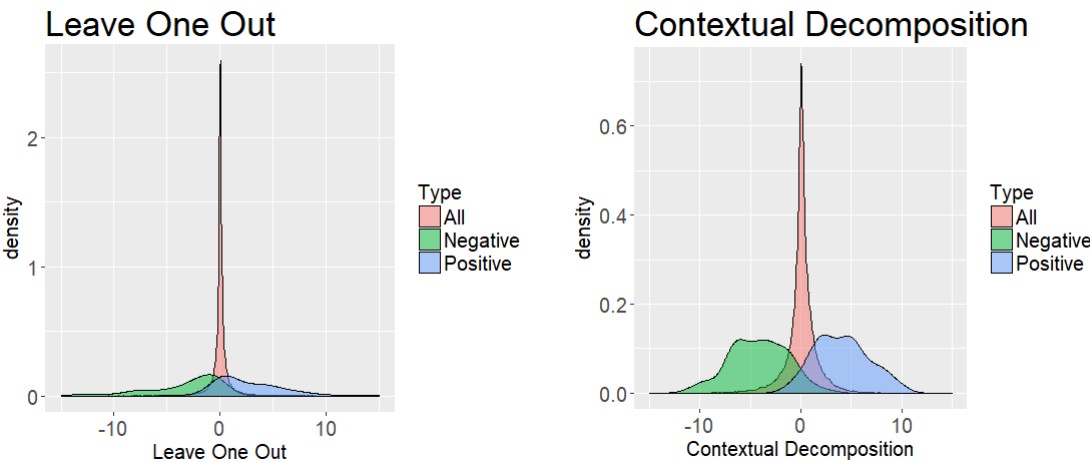

Figure 1: Distribution of scores for positive and negative negation coefficients relative to all interaction coefficients. Only leave one out and CD are capable of producing these interaction scores.

| not entertaining | not bad | very funny | entertaining | bad |
|---|---|---|---|---|
| not funny | never dull | well-put-together piece | intelligent | dull |
| not engaging | n't drag | entertaining romp | engaging | drag |
| never satisfactory | never fails | very good | satisfying | awful |
| not well | without sham | surprisingly sweet | admirable | tired |
| not fit | without missing | very well-written | funny | dreary |

Table 3: Nearest neighbours for selected unigrams and interactions using CD embeddings

cosine similarity metric. The results are qualitatively sensible for three different kinds of interactions: positive negation, negative negation and modification, as well as positive and negative words. Note that we for positive and negative words, we chose the positive/negative parts of the negations, in order to emphasize that CD can disentangle this composition.

## 5 CONCLUSION

In this paper, we have proposed contextual decomposition (CD), an algorithm for interpreting individual predictions made by LSTMs without modifying the underlying model. In both NLP and general applications of LSTMs, CD produces importance scores for words (single variables in general), phrases (several variables together) and word interactions (variable interactions). Using two sentiment analysis datasets for empirical validation, we first show that for information also produced by prior methods, such as word-level scores, our method compares favorably. More importantly, we then show that CD is capable of identifying phrases of varying sentiment, and extracting meaningful word (or variable) interactions. This movement beyond word-level importance is critical for understanding a model as complex and highly non-linear as LSTMs.

### ACKNOWLEDGMENTS

This research was started during a summer internship at Google Brain, and later supported by a postgraduate scholarship-doctoral from NSERC and a data science research award from Adobe. This work is partially supported by Center for Science of Information (CSoI), an NSF Science and

Technology Center, under grant agreement CCF-0939370, ONR grant N00014-16-1-2664 and ARO grant W911NF1710005.

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

| Attribution Method | Stanford Sentiment | Yelp Polarity |
|---|---|---|
| Gradient | 0.375 | 0.336 |
| Leave one out (Li et al., 2016) | 0.510 | 0.358 |
| Cell decomposition (Murdoch & Szlam, 2017) | 0.490 | 0.560 |
| Integrated gradients (Sundararajan et al., 2017) | 0.724 | 0.471 |
| Contextual decomposition | 0.758 | 0.520 |

Table 4: Correlation coefficients between logistic regression coefficients and extracted scores.

Kelvin Xu, Jimmy Ba, Ryan Kiros, Kyunghyun Cho, Aaron Courville, Ruslan Salakhudinov, Rich Zemel, and Yoshua Bengio. Show, attend and tell: Neural image caption generation with visual attention. In *International Conference on Machine Learning*, pp. 2048–2057, 2015.

Xiang Zhang, Junbo Zhao, and Yann LeCun. Character-level convolutional networks for text classification. In *Advances in neural information processing systems*, pp. 649–657, 2015.

# 6 APPENDIX

## 6.1 PLOTS

### 6.1.1 PLOTS FOR DISSENTING SUBPHRASES

We provide here the plots described in Section 4.3.

### 6.1.2 PLOTS FOR HIGH-LEVEL COMPOSITIONALITY

We provide here the plots referenced in Section 4.4.

### 6.1.3 LOGISTIC REGRESSION VERSUS EXTRACTED COEFFICIENTS SCATTERPLOTS

We provide here the scatterplots and correlations referenced in section 4.2.

## 6.2 GENERAL RECURSION FORMULA

We provide here the general recursion formula referenced in Section 3.2.1. The two cases that are considered is whether the current time step is during the phrase ($q \leq t \leq r$) or outside of the phrase ($t < q$ or $t > r$).

$$\beta_t^f = [L_\sigma(V_f\beta_{t-1}) + L_\sigma(b_f) + L_\sigma(V_f x_t)1_{q\leq t\leq r}] \odot \beta_{t-1}^c \tag{29}$$

$$\gamma_t^f = f_t \odot \gamma_{t-1}^c + [L_\sigma(V_f\gamma_{t-1}) + L_\sigma(V_f x_t)1_{t>q,t<r}] \odot \beta_{t-1}^c \tag{30}$$

$$\beta_t^u = L_\sigma(V_i\beta_{t-1}^c) \odot [L_{\tanh}(V_g\beta_{t-1}^c + L_{\tanh}(b_g)] + L_\sigma(b_i) \odot L_{\tanh}(V_g\beta_{t-1}^c) \tag{31}$$

$$+ [L_\sigma(W_i x_t) \odot [L_{\tanh}(W_g x_t) + L_{\tanh}(V_g\beta_{t-1}) + L_{\tanh}(b_g)] + L_\sigma(b_i) \odot L_{\tanh}(W_g x_t)]1_{q\leq t\leq r}$$

$$\gamma_t^u = L_\sigma(V_i\gamma_{t-1}) \odot g_t + i_t \odot L_{\tanh}(V_g\gamma_{t-1}) - L_\sigma(V_i\gamma_{t-1}) \odot L_{\tanh}(V_g\gamma_{t-1}) + L_\sigma(b_i) \odot L_{\tanh}(b_g)+ \tag{32}$$

$$+ [L_\sigma(W_i x_t) \odot [L_{\tanh}(W_g x_t) + L_{\tanh}(V_g\beta_{t-1}) + L_{\tanh}(b_g)] + L_\sigma(b_i) \odot L_{\tanh}(W_g x_t)]1_{t<q,t>r}$$

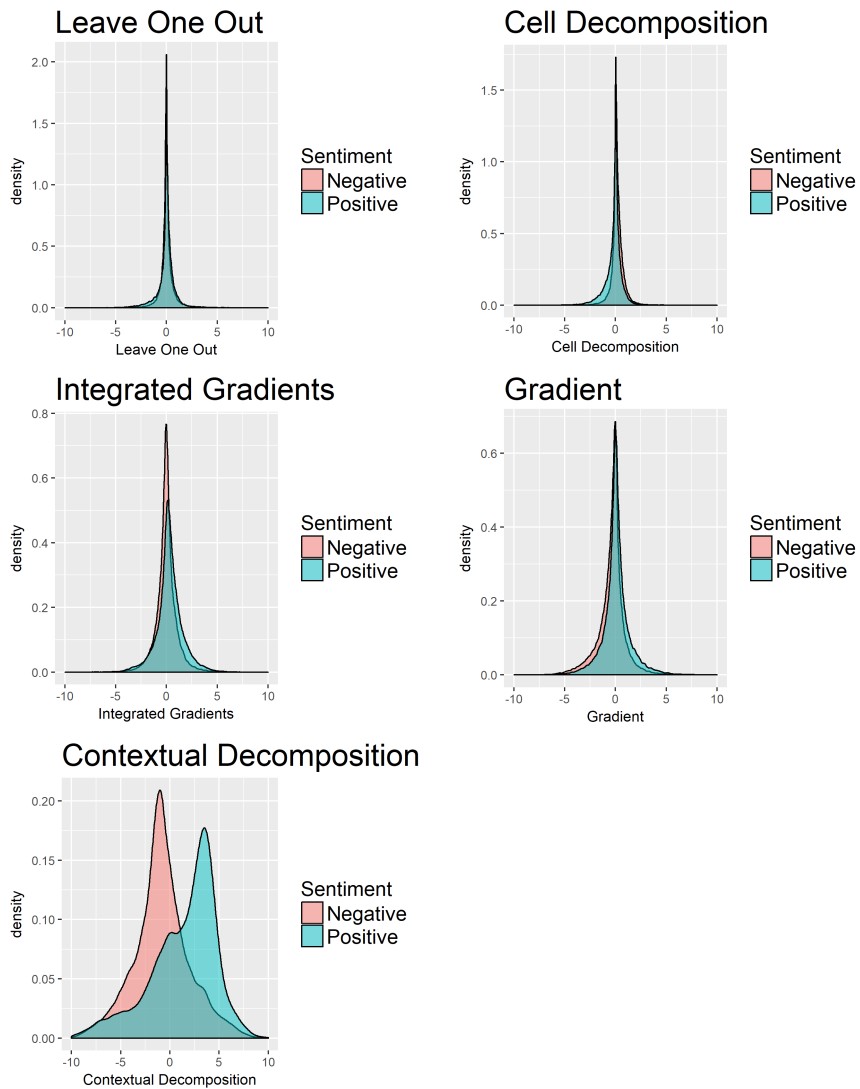

Figure 2: The distribution of attributions for positive (negative) sub-phrases contained within negative (positive) phrases of length at most five in the Yelp polarity dataset. The positive and negative distributions are nearly identical for all methods except CD, indicating an inability of prior methods to distinguish between positive and negative phrases when occurring in the context of a phrase of the opposite sentiment

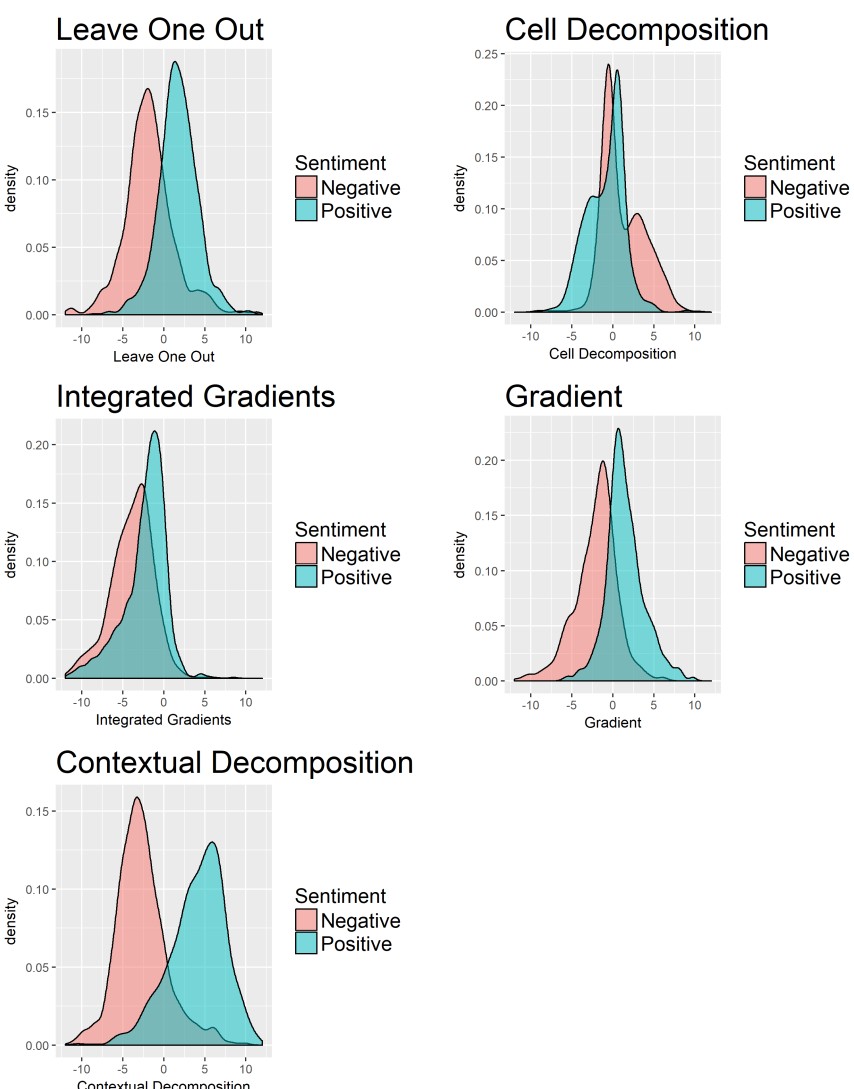

Figure 3: Distribution of positive and negative phrases, of length between one and two thirds of the full review, in SST. The positive and negative distributions are significantly more separate for CD than other methods, indicating that even at this coarse level of granularity, other methods still struggle.

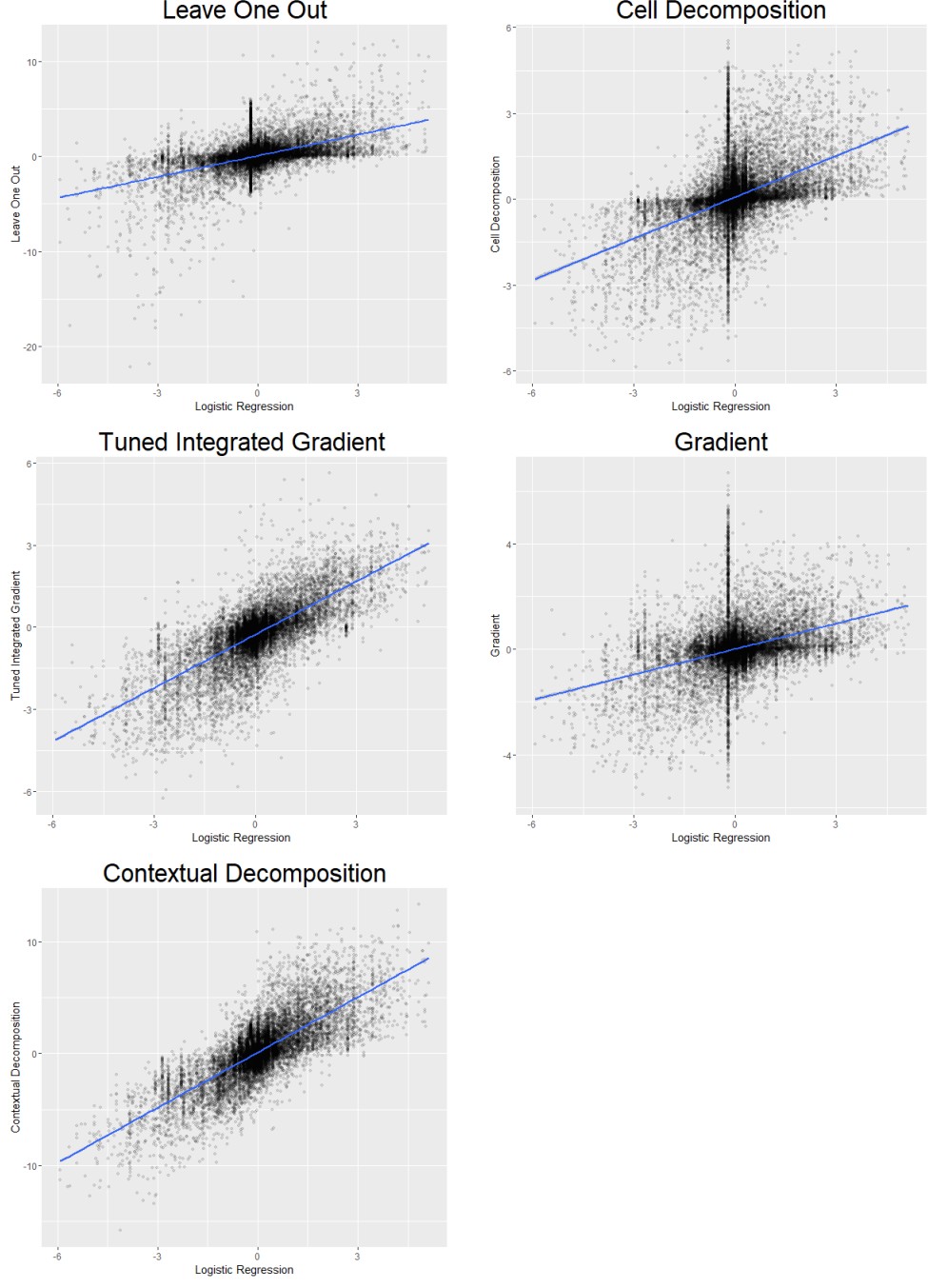

Figure 4: Logistic regression coefficients versus coefficients extracted from an LSTM on SST. We include a least squares regression line. Stronger linear relationships in the plots correspond to better interpretation techniques.

### 6.3 LIST OF WORDS USED TO IDENTIFY NEGATIONS

To search for negations, we used the following list of negation words: not, n't, lacks, nobody, nor, nothing, neither, never, none, nowhere, remotely

