# OpenReview forum: "Beyond Word Importance:  Contextual Decomposition to Extract Interactions from LSTMs"
_ICLR.cc/2018/Conference — Accept (Oral)_

### Official Review · AnonReviewer2 · 2017-11-27
**The paper proposes a decomposition of LSTM's gates allowing to interpret the contribution of each element in a sequence to the prediction and how their representations are combined.**

**Rating:** 7
**Confidence:** 4

**Review:**

In this paper, the authors propose a new LSTM variant that allows to produce interpretations by capturing the contributions of the words to the final prediction and the way their learned representations are combined in order to yield that prediction. They propose a new approach that they call Contextual Decomposition (CD). Their approach consists of disambiguating interaction between LSTM’s gates where gates are linearized so the products between them is over linear sums of contributions from different factors. The hidden and cell states are also decomposed in terms of contributions to the “phrase” in question and contributions from elements outside of the phrase. The motivation of the proposed decomposition using LSTMs is that the latter are powerful at capturing complex non-linear interactions, so, it would be useful to observe how these interactions are handled and to interpret the LSTM’s predictions. As the authors intention is to build a way of interpreting LSTMs output and not to increase the model’s accuracy, the empirical results illustrate the ability of their decomposition of giving a plausible interpretation to the elements of a sentence. They compare their method with different existing method by illustrating samples from the Yelp Polarity and SST datasets. They also show the ability of separating the distribution of CD scores related to positive and negative phrases on respectively Yelp Polarity and SST.

The proposed approach is potentially of great benefit as it is simple and elegant and could lead to new methods in the same direction of research. The sample illustrations, the scatter plots and the CD score distributions are helpful to asses the benefit of the proposed approach.

The writing could be improved as it contains parts where it leads to confusion. The details related to the linearization (section 3.2.2), the training (4.1) could be improved. In equation 25, it is not clear what  π_{i}^(-1) and x_{π_{i}} represent but the example in equation 26 makes it clearer. The section would be clearer if each index and notation used is explained explicitly.

(CD in known for Contrastive Divergence in the deep learning community. It would be better if Contextual Decomposition is not referred by CD.)

Training details are given in section 4.1 where the authors mention the integrated gradient baseline without mentioning the reference paper to it (however they do mention the reference paper at each of table 1 and 2). it would be clearer for the reader if the references are also mentioned in section 4.1 where integrated gradient is introduced. Along with the reference, a brief description of that baseline could be given.

The “Leave one out” baseline is never mentioned in text before section 4.4 (and tables 1 and 2). Neither the reference nor the description of the baseline are given. It would have been clearer to the reader if this had been the case.

Overall, the paper contribution is of great benefit. The quality of the paper could be improved if the above explanations and details are given explicitly in the text.

---

> ### Author Response · Authors · 2017-12-22
> **Response**
>
> Thanks for the detailed and thoughtful review. We've responded to some of your comments below.
>
> "In this paper, the authors propose a new LSTM variant that allows to produce interpretations by capturing the contributions of the words to the final prediction and the way their learned representations are combined in order to yield that prediction. "
> To clarify, we are not proposing a new neural architecture. Our new method, contextual decomposition, is an interpretation method for a standard LSTM. Given a trained LSTM, it can be applied without altering the underlying model, re-training, or any additional work. We think that this is more impactful than proposing a new architecture, as it doesn't force users to alter their model to get interpretations, nor to sacrifice the LSTM's predictive accuracy.
>
> This was a common misconception across reviewers, so we updated our abstract (lines 4-6), introduction (paragraph 2, lines 1-3) and conclusion (lines 1-2) to clarify this distinction.
>
> "The writing could be improved as it contains parts where it leads to confusion. The details related to the linearization (section 3.2.2), the training (4.1) could be improved. In equation 25, it is not clear what π_{i}^(-1) and x_{π_{i}} represent but the example in equation 26 makes it clearer. The section would be clearer if each index and notation used is explained explicitly."
> Thanks for pointing out these areas for improvement. We've expanded 3.2.2 to make it clearer and better motivate our notation, adding equation 26 and re-writing the subsequent paragraph. The x_{\pi_{i}} you note was actually a typo - it has been corrected to y_{\pi_{i}}.
>
> For 4.1, we split off the baseline portion of 4.1 into 4.1.3, which should make it clearer, while also addressing some of your concerns below around introducing, citing and describing baselines.
>
> "Training details are given in section 4.1 where the authors mention the integrated gradient baseline without mentioning the reference paper to it (however they do mention the reference paper at each of table 1 and 2). it would be clearer for the reader if the references are also mentioned in section 4.1 where integrated gradient is introduced. Along with the reference, a brief description of that baseline could be given. "
> Integrated gradients is now properly referenced. As discussed above, we added 4.1.3, which includes proper references, and refers the reader to the related work section for a description (We felt that reproducing descriptions of baselines in 4.1.3 would basically require replicating the related work paragraph).
>
> "The “Leave one out” baseline is never mentioned in text before section 4.4 (and tables 1 and 2). Neither the reference nor the description of the baseline are given. It would have been clearer to the reader if this had been the case."
> Thanks for pointing this out. We added a mention and citation of leave one out in the new baselines section, 4.1.3. Although the paper and method were discussed in related work (paragraph 1, lines 5-7), we added a reference to the name “leave one out” for clarity.

---

### Official Review · AnonReviewer1 · 2017-11-27
**A promising paper on how interpretability can boost the performance LSTMs**

**Rating:** 7
**Confidence:** 2

**Review:**

The authors address the problem of making LSTMs more interpretable via the contextual decomposition of the state vectors. By linearizing the updates in the recurrent network, the proposed scheme allows one to extract word importance information directly from the gating dynamics and infer word-to-word interactions.

The problem of making neural networks more understandable is important in general. For NLP, this relates to the ability of capturing phrase features that go beyond single-word importance scores. A nice contribution of the paper is to show that this can highly improve classification performance on the task of sentiment analysis. However, the author could have spent some more time in explaining the technical consequences of the proposed linear approximation. For example, why is the linear approximation always good? And what is the performance loss compared to a fully nonlinear network?

The experimental results suggest that the algorithm can outperform state-of-the-art methods on various tasks.

Some questions:
- is any of the existing algorithms used for the comparison supposed to capture interactions between words and phrases? If not, why is the proposed algorithm compared to them on interaction related tasks?
- why the output of the algorithms is compared with the logistic regression score? May the fact that logistic regression is a linear model be linked to the good performance of the proposed method? Would it be possible to compare the output of the algorithms with human given scores on a small subset of words?
- the recent success of LSTMs is often associated with their ability to learn complex and non-linear relationships but the proposed method is based on the linearization of the network. How can the algorithm be able to capture non-linear interactions? What is the difference between the proposed model and a simple linear model?

---

> ### Author Response · Authors · 2017-12-22
> **Response**
>
> Thank you for your helpful comments.
> "A nice contribution of the paper is to show that this can highly improve classification performance on the task of sentiment analysis. "
> We actually do something different than what this implies. CD is an algorithm for producing interpretations of LSTM predictions. In particular, given a trained LSTM, it produces interpretations for individual predictions without modifying the LSTM's architecture in any way, leaving the predictions, and accuracy, unchanged. The purpose of our evaluation is not to improve predictive performance but rather to demonstrate that these importance scores accurately reflect the LSTMs dynamics (e.g. negation, compositionality) and, in particular, do so better than prior methods. We have updated our abstract (lines 4-6), introduction (paragraph 2, lines 1-3), conclusion (lines 1-2), and added equation 10 in order to avoid similar misunderstandings by future readers.
> "However, the author could have spent some more time in explaining the technical consequences of the proposed linear approximation. For example, why is the linear approximation always good? And what is the performance loss compared to a fully nonlinear network? "
> Our linearization is not an approximation, it is exact. CD produces an exact decomposition of the values fed into the LSTM’s final softmax into a sum of two terms: contributions resulting solely from the specified phrase, and others. Mathematically, this is shown in the newly added equation 10. Moreover, CD is used for interpretation of the original LSTM, not as a separate prediction algorithm, so that there is no performance loss.
> "- is any of the existing algorithms used for the comparison supposed to capture interactions between words and phrases? If not, why is the proposed algorithm compared to them on interaction related tasks?"
> This is a great question, we assume you’re referencing the finding in 4.5 that CD can extract negations. To the best of our knowledge, no prior algorithm has made the claim of being able to extract interactions from LSTMs. Our ability to do so is a significant contribution. Although not previously discussed, the leave one out method can be adapted to produce an interaction value, which we report in figure 1. However, the produced interactions don't seem to contain much information, perhaps explaining why they were not included in the original paper. Nonetheless, leave one out is the only baseline we are aware of, so we thought it important to report them for comparison.
> "why the output of the algorithms is compared with the logistic regression score? May the fact that logistic regression is a linear model be linked to the good performance of the proposed method?"
> I assume you're referencing 4.2 here. When the underlying model is sufficiently accurate (which it is in our case), logistic regression coefficients are generally viewed to provide qualitatively sensible importance scores. In other words, the ordering provided by the coefficients generally lines up very well with what humans qualitatively view as important. Thus, a sensible check for the behaviour of an interpretation algorithm is whether or not it can recover qualitatively similar coefficients, as measured by correlation.
>
> To elaborate, if a logistic regression coefficient is very positive, such as for “terrific”, we would expect the word importance score extracted from an LSTM to also be very positive. Similarly, if the logistic regression coefficient is zero, such as for “the”, we would expect the LSTM's word importance to be quite small. We do not expect these relationships to be perfect, but the fact that they are reasonably strong supports our claim that our method produces comparable or superior word importance scores.
>
> “Would it be possible to compare the output of the algorithms with human given scores on a small subset of words?”
> The problem of running human experiments to validate interpretations is an interesting and active research area. However, running such experiments is a substantive endeavour, which unfortunately puts it outside the scope of this paper. We do agree that it would be an exciting prospect for future work, though. Nonetheless, as discussed above, we do believe that our approach provides valuable information, if not as much as a full human experiment.

---

> > ### Author Response · Authors · 2017-12-22
> > **Response p2**
> >
> > "- the recent success of LSTMs is often associated with their ability to learn complex and non-linear relationships but the proposed method is based on the linearization of the network. How can the algorithm be able to capture non-linear interactions? What is the difference between the proposed model and a simple linear model? "
> > We hope that our earlier comments resolve this question. In particular, our proposed method is not a separate prediction method, but rather an interpretation method for a standard LSTM. In response to this confusion, we have updated our abstract (lines 4-6), introduction (paragraph 2, lines 1-3), conclusion (lines 1-2), and added equation 10.

---

### Official Review · AnonReviewer3 · 2017-11-27
**An interesting fine-grained model which lacks of a proper comparison with the state of the art**

**Rating:** 7
**Confidence:** 3

**Review:**

This article aims at understanding the role played by the different words in a sentence, taking into account their order in the sentence. In sentiment analysis for instance, this capacity is critical to model properly negation.
As state-of-the-art approaches rely on LSTM, the authors want to understand which information comes from which gate. After a short remainder regarding LSTM, the authors propose a framework to disambiguate interactions between gates. In order to obtain an analytic formulation of the decomposition, the authors propose to linearize activation functions in the network.
In the experiment section, authors compare themselves to a standard logistic regression (based on a bag of words representation). They also check the unigram sentiment scores (without context).
The main issue consists in modeling the dynamics inside a sentence (when a negation or a 'used to be' reverses the sentiment). The proposed approach works fine on selected samples.


The related work section is entirely focused on deep learning while the experiment section is dedicated to sentiment analysis. This section should be rebalanced. Even if the authors claim that their approach is general, they also show that it fits well the sentiment analysis task in particular.

On top of that, a lot of fine-grained sentiment analysis tools has been developed outside deep-learning: the authors should refer to those works.

Finally, authors should provide some quantitative analysis on sentiment classification: a lot of standard benchmarks are widely use in the literature and we need to see how the proposed method performs with respect to the state-of-the-art.


Given the chosen tasks, this work should be compared to the beermind system:
http://deepx.ucsd.edu/#/home/beermind
and the associated publication
http://arxiv.org/pdf/1511.03683.pdf

---

> ### Author Response · Authors · 2017-12-22
> **Response**
>
> Thank you for your helpful comments. As you'll see below, they lead to a meaningful improvement in the framing of our paper.
>
> The problem we are solving is not extracting interactions, in a general sense, from text data, nor is it predicting sentiment. The problem we are solving is, for a given, trained, LSTM, explaining individual predictions being made, without modifying the architecture. This is an important distinction, which informs what denotes related work, and what methods we compare against. It is also one that was not entirely clear in the original version, and we so we updated our abstract (lines 4-6), introduction (paragraph 2, lines 1-3), conclusion (lines 1-2) and added equation 10 to better express this. When framed in this way, we believe that our baselines are the correct ones to demonstrate the novelty of our results.
>
> "In the experiment section, authors compare themselves to a standard logistic regression (based on a bag of words representation). They also check the unigram sentiment scores (without context).
> The main issue consists in modeling the dynamics inside a sentence (when a negation or a 'used to be' reverses the sentiment). The proposed approach works fine on selected samples."
>
> To clarify, only section 4.2 compares against a logistic regression, and deals with solely unigram sentiment scores. Sections 4.3-4.6 do not involve logistic regression, and deal with general n-grams and interactions.
>
> It is worth noting that we were very careful to not rely on "selected samples", i.e. cherry-picking, as our primary means of validation. Rather, we provide anecdotes to motivate searches across the full dataset for different types of compositionality, with each of 4.3, 4.4 and 4.5 involving different criteria, such as negation. For each of these different instances, we ultimately base our conclusions on the distributions of importance scores extracted from our LSTM across all phrases/reviews containing each kind of compositionality. These distributions can be found in figures 1-4 and provide, in our opinion, a more compelling case.
>
> "The related work section is entirely focused on deep learning while the experiment section is dedicated to sentiment analysis. This section should be rebalanced. Even if the authors claim that their approach is general, they also show that it fits well the sentiment analysis task in particular."
>
> The primary contribution of this paper is an algorithm for interpreting predictions made by LSTMs, not improved prediction performance on sentiment analysis. Consequently, the related work section focuses on prior work in interpreting deep learning algorithms, particularly LSTMs. In our experiment section, we fit a single LSTM per dataset and analyse the behaviour of the LSTM interpretations produced by CD, along with four interpretation baselines, in different settings. The LSTMs are fit using standard procedures, and we make no claims of state of the art predictive performance from our model.
>
> "Finally, authors should provide some quantitative analysis on sentiment classification: a lot of standard benchmarks are widely use in the literature and we need to see how the proposed method performs with respect to the state-of-the-art."
>
> To be clear, we assume that when you refer to benchmarks, and ask for performance with respect to state-of-the-art, you are referring to predictive accuracy. We do not claim to be state of the art in terms of predictive accuracy. In fact, as we note in 4.1.1 and 4.1.2, our models follow implementations of baselines used for predictive accuracy in prior papers.
>
> Rather, what we do claim is state of the art for interpreting predictions made by an LSTM. To justify this claim, we compare against four LSTM interpretation benchmarks across four different evaluation settings. Given the focus of our paper, we thought these were the most relevant comparisons.
>
> As we mentioned above, we've updated our abstract (lines 4-6), introduction (paragraph 2, lines 1-3), conclusion (lines 1-2) to clarify this distinction.
>
> "Given the chosen tasks, this work should be compared to the beermind system: http://deepx.ucsd.edu/#/home/beermind and the associated publication http://arxiv.org/pdf/1511.03683.pdf"
>
> Thanks for the pointer, this paper was a very interesting read. It seems that the focus is primarily on generating reviews for a given user/item pair, and secondarily on predicting sentiment. Given that our paper is focused on interpreting LSTMs, not on generating reviews or predictive performance for sentiment analysis, we are unsure what a meaningful, relevant comparison would look like.

---

> > ### Comment · AnonReviewer3 · 2017-12-29
> > **Why baselines are performing so poorly?**
> >
> > The authors focus on the interpretation of an existing LSTM. They use the sentiment classification task to illustrate the behavior of their approach. The work done on LSTM is interesting, explaining how the latent representation at t has been built from x_t and h_{t-1}.
> > Regarding the chosen task, my opinion is more lukewarm: results obtained by the other methods are completly awful. They are so far from the ground truth that it is difficult to consider them as baselines (in fact, on the proposed exemples, baselines give results at the opposite from what is expected). In that sense, I still would like to compare CD interpretation with real method adapted to fine grained sentiment classification.
> > As the author propose a quantitative analysis regarding their approach, we would like to compare those figures with the state of the art.

---

> > > ### Author Response · Authors · 2017-12-31
> > > **This is a young field, the baselines aren't very good. Our method improves on them.**
> > >
> > > Thanks for your thoughtful response.
> > >
> > > "Regarding the chosen task, my opinion is more lukewarm: results obtained by the other methods are completly awful. They are so far from the ground truth that it is difficult to consider them as baselines (in fact, on the proposed exemples, baselines give results at the opposite from what is expected)."
> > >
> > > We agree with you that it is surprising how poorly prior methods perform, and also that CD provides significant improvements over multiple state of the art baselines. It is worth noting that work in interpreting predictions made by LSTMs is still very young. To the best of our knowledge, the first paper in this area, was presented one year ago at ICLR 2017 [1]. Moreover, concurrent ICLR 2018 submissions [2][3] have also presented evidence that existing interpretation methods for neural networks can perform quite poorly. This is to say, the problem of interpreting neural networks in general, and LSTMs in particular, is far from solved. Hopefully this makes the shortcomings of existing work slightly less surprising. We believe that we have provided strong evidence that our method has made substantial progress in an area where it is clearly needed.
> > >
> > > "In that sense, I still would like to compare CD interpretation with real method adapted to fine grained sentiment classification.
> > > As the author propose a quantitative analysis regarding their approach, we would like to compare those figures with the state of the art."
> > >
> > > The problem we are trying to solve is that of producing explanations for individual predictions made by an LSTM. To the best of our knowledge, the methods we compare against are state of the art for solving this problem. If you know of any additional methods for solving this problem, we would appreciate if you could point them out to us.
> > >
> > > Moreover, this work serves as the first paper to extract interactions from LSTMs (as presented in 4.5), an important task for which there is no prior work. While we use leave one out as a baseline, it was never claimed to perform this task, and struggles accordingly. Again, we would appreciate pointers to any additional methods.
> > >
> > > As a side note, by "real method", we assume you mean a method which is state of the art for predictive accuracy. It is worth noting that, for Stanford Sentiment Treebank, which is still actively published on, the state of the art is dominated by deep learning, largely LSTMs with various modifications. For instance, here's a recent paper from openAI [4] (see figure 2 on page 3), co-authored by Ilya Sutskever.
> > >
> > > [1] https://arxiv.org/abs/1702.02540 - WJ Murdoch, A Szlam, Automatic Rule Extraction from Long Short Term Memory Networks
> > > [2] https://openreview.net/forum?id=H1xJjlbAZ - anon., Interpretation of neural networks is fragile
> > > [3] https://openreview.net/forum?id=r1Oen--RW - anon., The (un)reliability of salience methods
> > > [4] https://arxiv.org/pdf/1704.01444.pdf - A Radford, R Jozefowicz, I Sutskever, Learning to Generate Reviews and Discovering Sentiment

---

> > > > ### Comment · AnonReviewer3 · 2018-01-12
> > > > **Thank you for all the comments**
> > > >
> > > > Explanations are convincing, I revise my rating.

---

> > > > > ### Author Response · Authors · 2018-01-13
> > > > > **Thanks**
> > > > >
> > > > > Thanks for engaging in a helpful discussion!

---

### Public Comment · (anonymous) · 2018-02-18
**Prior work on explaining LSTM predictions**

Here is another relevant prior work: http://aclweb.org/anthology/W17-5221

---

### Public Comment · ~Guillermo_Valle_Perez1 · 2022-05-08
**Issue understanding how the CD score is interpreted**

As far as I understand, the CD score is a measure of how relevant an input is for the output. However, in the experimental section, it is interpreted as a measure of sentiment of a section of the input.
I don't understand the rationale behind this. In a phrase like "not good", I would expect both "not" and "good" to get high relevance scores (as changing either would affect the output (i.e. both "good" and "not bad" are positive). However, "not" has negative sentiment and "good" has positive sentiment. I find it confusing that CD would assign a low score to "good" in "not good" as the output of the prediction should depend highly on "good", shouldn't it?

I also find it weird that CD behaves in the way shown, given that in the equations the parts which include interactions from the relevant and irrelevant parts, are counted as relevant. I would expect the positive output of "not bad" to come from those interactions, and so I would expect the CD score of both "not" and "bad" to be high for "not bad". Hmm, what am In missunderstading?

---

> ### Author Response · Authors · 2022-05-08
> **Response (but please move to email/don't resuscitate 5 year old threads :) )**
>
> Hey Guillermo!
>
> These reviews are almost five years old, I'll respond here but let's move this to email? My address is on the paper.
>
> "As far as I understand, the CD score is a measure of how relevant an input is for the output. However, in the experimental section, it is interpreted as a measure of sentiment of a section of the input."
>
> In our case, we are doing sentiment analysis, so the output is sentiment. So, the assumption is that something is (or should be) relevant to the output if it has positive/negative sentiment.
>
> "In a phrase like "not good", I would expect both "not" and "good" to get high relevance scores (as changing either would affect the output (i.e. both "good" and "not bad" are positive)."
>
> Basically the whole point of this paper is that "not" should have a low relevance score in "not good". Essentially, there are 3 pieces of information an accurate model should learn -
> 1) that having "not", on it's own, in a sentence doesn't impact the sentiment much,
> 2) That "good" increases the sentiment, on it's own, and
> 3) that "not good" decreases the sentiment.
> If you're limited to word importance scores, you could make realistic arguments that "not" and "good" should be any combination of positive/negative (and in practice, different word importance scores will give you different combinations), making their outputs essentially useless. If you use phrase importance scores (which was new when the CD paper came out), you can get all 3.
>
> The crux of this paper (which I feel is still relevant today) is that to explain non-linear models you need non-linear explanations, i.e. that you should be able to capture all 3 pieces mentioned above. And feature importance scores are linear, so can only capture 2 of those 3 (and they don't tell you which 2!).
>
> (As a side note, blacking out a word/phrase and measuring the change on a model's prediction is an approach for feature importance that has been studied, and that we compare against (Leave one out, Li et. al 2016), and that has generally been discarded in more recent work)
>
> Also see the talk I gave at ICLR 2018, where I go into this in detail (starts around 1:05: https://www.facebook.com/iclr.cc/videos/2127071060655282)
>
> "I also find it weird that CD behaves in the way shown, given that in the equations the parts which include interactions from the relevant and irrelevant parts, are counted as relevant."
>
> You are mistaken - the interactions between relevant and irrelevant are included in the irrelevant parts, but relevant. To be precise, in equation (14), Lσ(Vf γt−1)  β^c_{t−1} is one of those terms, and in equation (17) Lσ(Viγt−1)  g_t + i_t  Ltanh(Vgγt−1) (the \beta's are in g_t, i_t).
>
> Also see the top of page 4, "Terms are determined to derive solely from the specified phrase if they involve products from some combination of β_{t−1}, β^c_{t−1}, xt and bi or bg (but not both)".
>
> Hope this helps!

---

### Decision · Program_Chairs · 2018-01-29
**ICLR 2018 Conference Acceptance Decision**

**Decision:**

Accept (Oral)

**Comment:**

Very solid paper exploring an interpretation of LSTMs.
good reviewss